# Two-tiered enforcement of high-fidelity DNA ligation

Percy P. Tumbale[1,3], Thomas J. Jurkiw [2,3], Matthew J. Schellenberg [1], Amanda A. Riccio[1], Patrick J O'Brien [2]* & R. Scott Williams [1]*

DNA ligases catalyze the joining of DNA strands to complete DNA replication, recombination and repair transactions. To protect the integrity of the genome, DNA ligase 1 (LIG1) discriminates against DNA junctions harboring mutagenic 3'-DNA mismatches or oxidative DNA damage, but how such high-fidelity ligation is enforced is unknown. Here, X-ray structures and kinetic analyses of LIG1 complexes with undamaged and oxidatively damaged DNA unveil that LIG1 employs $Mg^{2+}$-reinforced DNA binding to validate DNA base pairing during the adenylyl transfer and nick-sealing ligation reaction steps. Our results support a model whereby LIG1 fidelity is governed by a high-fidelity (HiFi) interface between LIG1, $Mg^{2+}$, and the DNA substrate that tunes the enzyme to release pro-mutagenic DNA nicks. In a second tier of protection, LIG1 activity is surveilled by Aprataxin (APTX), which suppresses mutagenic and abortive ligation at sites of oxidative DNA damage.

[1] Genome Integrity and Structural Biology Laboratory, National Institute of Environmental Health Sciences, US National Institutes of Health, Department of Health and Human Services, 111 TW Alexander Drive, Research Triangle Park, NC 27709, USA. [2] Biological Chemistry, University of Michigan, 1150 W Medical Center Drive Ann Arbor, Ann Arbor, MI 48109, USA. [3] These authors contributed equally: Percy P. Tumbale, Thomas J. Jurkiw. *email: pjobrien@umich.edu; williamsrs@niehs.nih.gov

The maintenance of genomic integrity requires high fidelity in all DNA transactions, including DNA replication, recombination, and repair. Extensive efforts have established how DNA polymerases discriminate against the incorporation of incorrect nucleotides to varying degrees on the basis of base pairing (correct versus incorrect) and/or sugar identity (deoxyribonucleotides versus ribonucleotides). For replicative polymerases, the overall fidelity is dictated by intrinsic selectivity for the correct nucleotides, exonucleolytic proofreading, and post-replicative repair pathways such as mismatch DNA repair and ribonucleotide excision repair[1,2]. It is known that DNA ligases, the key enzymes responsible for completing replication and repair pathways, also contribute to overall fidelity[3–5]. However, the underlying mechanisms that regulate high fidelity ligation are poorly understood.

X-ray crystal structures of the three human DNA ligases (LIG1, LIG3, and LIG4) in complex with DNA have revealed a conserved ligase three-domain core architecture that encircles the DNA nick[6–9], induces partial unwinding and alignment of the 3′- and 5′-DNA ends, and distorts the 3′-OH strand into a C3′-endo sugar conformation to adopt an A-form like geometry[7–9]. Despite these similarities, DNA ligases display different fidelity profiles and are either tuned for high-fidelity ligation (e.g., human DNA ligase 1 homologs) or have relaxed stringency for correct DNA base pairing during DNA nick sealing (e.g., DNA ligase 4)[5,10–12]. Of the three human DNA ligases, LIG1 is established as the replicative ligase, completing ligation of the >50 million ligation events during DNA replication in addition to ligation events during DNA repair. LIG1 is essential for embryonic development[13] and LIG1 mutants are linked to immunodeficiencies[14–16]. LIG1 discriminates against 3′ mismatches[17] and DNA damage, including oxidative DNA damage incorporated by DNA polymerases[18]. Discrimination by DNA ligases can impair progression of DNA repair when oxidized nucleotides (8oxo-dG, 8oxo-deoxyguanosine) are inserted by DNA repair polymerases[18–20]. Moreover, the toxicity of anti-cancer MutT homolog 1 (MTH1) inhibitors that increase concentrations of oxidatively damaged nucleotide pools is hypothesized to act in part by promoting DNA repair polymerase incorporation of oxidized nucleotides and subsequent ligation by DNA ligases[21]. The extent to which discrimination by DNA ligases counteracts polymerase-promoted mutagenesis remains incompletely defined.

Human DNA ligases utilize the energy of ATP and catalyze phosphodiester bond formation to seal breaks in the DNA backbone (Fig. 1a). In the first of three chemical steps, an active site lysine (K568 in LIG1) is adenylylated by ATP, releasing pyrophosphate. After a conformational change, the enzyme binds a DNA nick and catalyzes adenylyl transfer to the 5′ phosphate of the nick to generate a 5′-5′ phosphoanhydride intermediate. In the final step, LIG1 catalyzes attack of the 3′ hydroxyl to seal the phosphodiester backbone and release AMP. The discrimination between proper and improper nicks could occur in either step 2 or step 3. Previous studies with an NAD-dependent enzyme *Escherichia coli* DNA ligase (LigA) provided evidence for discrimination in both steps[22]. However, appropriate molecular models that define the basis of DNA ligase fidelity have been lacking.

In the context of 3′-blocked ends, LIG1 has been shown to release the AMP–DNA species in a process referred to as abortive ligation[23–26]. Abortive ligation on damaged nicks could provide additional time for other DNA repair pathways to operate. However, it is necessary for the 5′-AMP group to be removed to restore a 5′-phosphate for ligation after 3′-proofreading and gap filling. Aprataxin (APTX) is a hydrolase that has been shown to catalyze the hydrolysis of 5′–5′ adenylylated DNA and RNA[12,23,24] and *APTX* mutations cause the neurodegenerative

disease Ataxia with Oculomotor Apraxia 1 (AOA1)[5,12,27,28]. LIG1 and APTX may together influence the overall fidelity of DNA ligation, but it is not established to what extent the activities of LIG1 and APTX might be coordinated.

In this work, we establish a foundation for understanding high-fidelity DNA ligation by human LIG1. We report a series of high-resolution structures of wild-type and mutant LIG1 bound to either correctly base-paired or damaged DNA. Unexpectedly, we found a conserved $Mg^{2+}$-binding site that forms a junction between DNA ligase domains and the 3′-strand of DNA. This led us to investigate the contribution of this structural motif to catalysis. We find that this site is dispensable for catalysis of step 2 and step 3, but it is critical for discrimination against incorrect base pairs at the 3′-end of the nick. Steady-state and pre-steady-state kinetic analysis reveals that this high-fidelity (HiFi) $Mg^{2+}$ architecture introduces strain that rejects improperly adenylylated breaks, resulting in abortive ligation. The structure of LIG1 bound to oxidatively damaged substrate highlights conformational changes in the active site that impair catalysis. We further demonstrate that APTX provides another tier of damaged DNA substrate discrimination by preferentially intercepting and hydrolyzing AMP–DNA intermediates that contain improper 3′ termini. Conservation of the HiFi $Mg^{2+}$-binding site in LIG1 homologs across species points to the importance of high-fidelity DNA ligation, but also reveals intriguing differences from LIG3 and LIG4, which lack an equivalent $Mg^{2+}$-binding motif. These structural differences may be key to the different specialization of these paralogous DNA ligases.

## Results

**Metal-reinforced DNA substrate engagement in human LIG1.** To define the molecular basis of LIG1 substrate recognition and fidelity, we sought protein–DNA crystallization conditions suitable for high-resolution structural determination. We evaluated 16 different nicked DNA duplexes with varying duplex length and nick location and formed complexes with two LIG1 constructs (aa 232–919 or 262–904, Fig. 1b). Sparse matrix screens covered > 36,000 conditions, ultimately identifying orthorhombic crystal forms of LIG1 that diffracted to high resolution (Table 1). Structure determination with a 3′ dideoxycytidine (ddC)-terminated substrate facilitated trapping of the LIG1 complex as the adenylylated–DNA intermediate, poised for nick sealing (step 3). Consistent with previous LIG1 structures determined at moderate resolution (3.0 Å)[7], the enzyme completely envelopes the DNA with the DNA-binding domain (DBD), adenylylation domain (AdD), and oligonucleotide-binding domain (OBD) encircling the nicked DNA (Fig. 1c).

The location of four coordinated $Mg^{2+}$ ions (Sites 1–4, Fig. 1d) at the protein–DNA interface were clearly delineated when 200 mM $Mg^{2+}$ was present in the cryoprotectant solution. Site 1 is coordinated at the three-way juncture of the AdD, DBD, and the DNA phosphodiester backbone (Fig. 2a). For reasons further elaborated below we refer to this cation as the high-fidelity $Mg^{2+}$ ($Mg^{HiFi}$). A second $Mg^{2+}$ (Site 2, $Mg^{cat}$) is coordinated at the active site where it presumably acts as the catalytic metal ion. Two additional sites (Site 3 and Site 4) are coordinated by protein ligands Glu346 (Site 3; Fig. 2a) and Glu804/Gly870 (Site 4; Fig. 1d).

The previous LIG1 structure determined in the presence of $Mg^{2+}$ did not provide interpretable density for bound $Mg^{2+}$ ions[7]. Therefore, the configuration of the catalytic $Mg^{2+}$ ($Mg^{cat}$) and the bound adenylylated intermediate provides important new insights into metal-dependent phosphodiester bond formation by ATP-dependent mammalian DNA ligases. The $Mg^{cat}$ metal-binding site has a slightly distorted coordination geometry, with

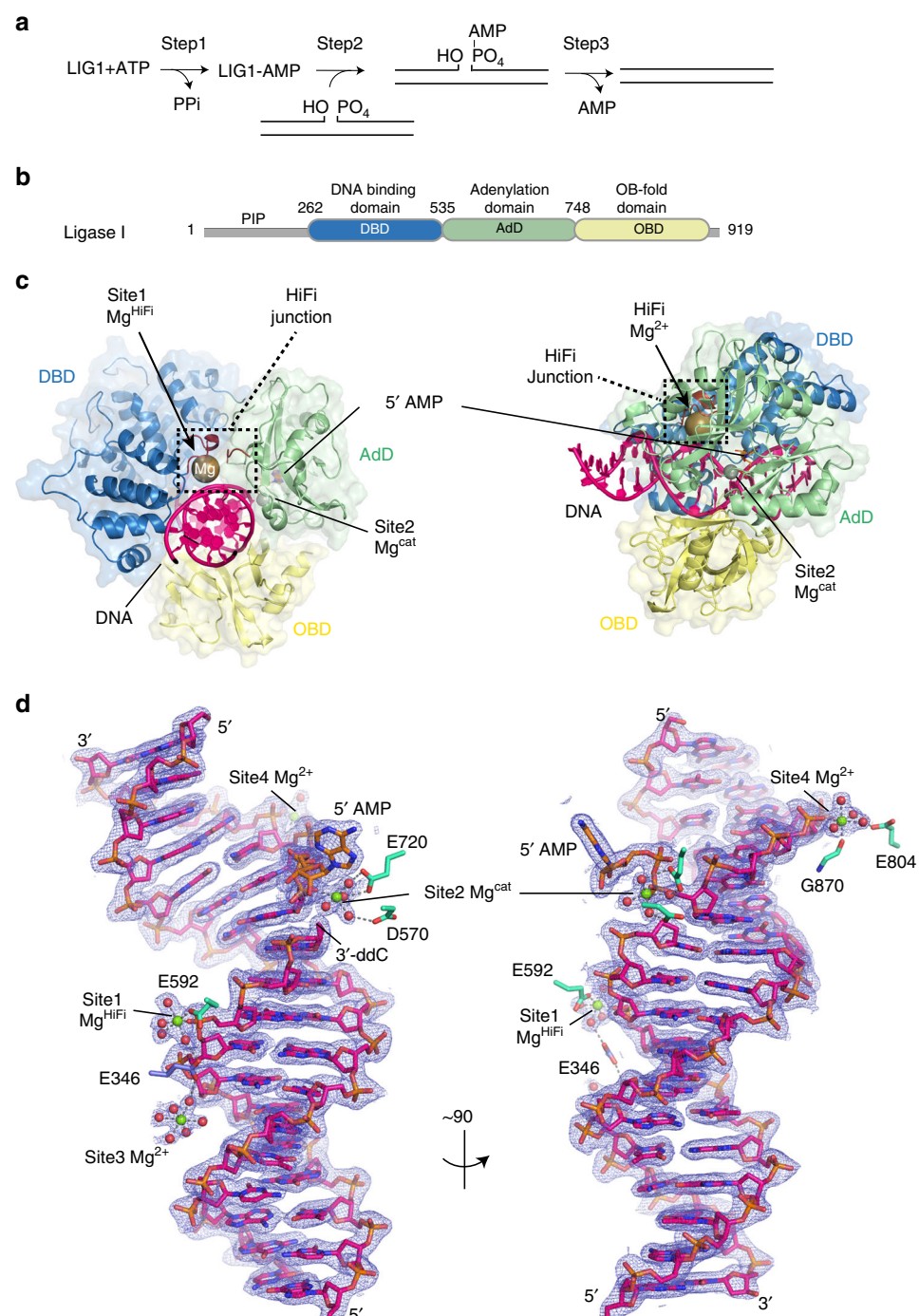

**Fig. 1** X-ray structure of LIG1•Mg$^{2+}$•adenylylated nicked DNA complex. **a** The three step ATP-dependent DNA ligation reaction. **b** Domain structure of human LIG1. The N-terminal domain (gray) contains a nuclear localization signal and a PIP box motif that mediates interactions with PCNA. The DNA-binding domain (DBD, blue), Adenylation domain (AdD, green), and OB-fold domain (OBD, yellow) comprise the catalytic core, mapping to residues 262–909. **c** X-ray structure of LIG1•Mg$^{2+}$•AMP–DNA complex. The DBD (blue), AdD (green), and OBD (yellow) domains encircle the nicked DNA substrate (pink). The 5′-phosphate of the nick is adenylylated (orange) and a Mg$^{2+}$ (gray) are bound in the active site. The Mg$^{HiFi}$ (brown) is bound in the HiFi junction highlighted in a dash-line box. **d** Final sigma-A weighted 2Fo-Fc electron density contoured at 1σ around the DNA and magnesium ions.

five of the six octahedral ligands present. The DNA 3′-OH is absent from this structure, but based on positioning of the deoxyribose ring, the 3′-OH is predicted to provide the sixth Mg$^{cat}$ ligand (Fig. 2b). Three stringently conserved acidic AdD residues (D570, E621, and E720)[7,29] act as outer sphere ligands by coordinating four water molecules around Mg$^{cat}$, whereas a

5′-phosphate oxygen mediates inner sphere ligand contacts to Mg$^{cat}$. Mg$^{cat}$ is positioned to stabilize the pentavalent transition state during the nick-sealing reaction in concert with R589, which forms a salt bridge to the 5′-phosphate and the bridging oxygen of the adenylylated intermediate. Two additional salt-bridging interactions are formed by K568 and K744 to the AMP

**Table 1 X-ray data collection and refinement statistics.**

|  | LIG1 WT 3′ddC 200 mM Mg$^{2+}$ | LIG1 WT 3′ddC 2 mM Mg$^{2+}$ + | LIG1 EE/AA 3′ddC 200 mM Mg$^{2+}$ | Lig1 WT 3′ OH EDTA | Lig1 EE/AA 3′ OH EDTA | Lig1 EE/AA 3′-8OG EDTA | Lig1 E592R 3′ OH EDTA |
|---|---|---|---|---|---|---|---|
| PDB entry ID | 6P09 | 6P0A | 6P0B | 6P0C | 6P0D | 6P0E | 6Q1V |
| **Data collection** | | | | | | | |
| Space group | P2$_1$2$_1$2$_1$ | P2$_1$2$_1$2$_1$ | P2$_1$2$_1$2$_1$ | P2$_1$2$_1$2$_1$ | P2$_1$2$_1$2$_1$ | P 2$_1$ 2$_1$ 2$_1$ | P 2$_1$ 2$_1$ 2$_1$ |
| *Cell dimensions:* | | | | | | | |
| *a, b, c* (Å) | 72.04 101.08 115.45 | 62.64, 110.85, 115.53 | 71.87 101.40 115.61 | 71.46 102.38 115.87 | 71.79 101.23 115.37 | 71.68 100.94 115.36 | 71.49 101.17, 115.29 |
| *α, β, γ* (°) | 90, 90, 90 | 90, 90, 90 | 90, 90, 90 | 90, 90, 90 | 90, 90, 90 | 90, 90, 90 | 90, 90, 90 |
| Resolution (Å) | 50–2.05 (2.12–2.05) | 50–2.05 (2.12–2.05) | 50–2.20 (2.28–2.20) | 50–1.55 (1.61–1.55) | 50–1.75 (1.81–1.75) | 50–1.85 (1.92–1.85) | 50–1.85 (1.92–1.85) |
| $R_{sym}$ | 0.098 (1.41) | 0.141 (1.45) | 0.107 (1.24) | 0.110 (1.93) | 0.102 (1.02) | 0.112 (1.53) | 0.084 (1.61) |
| $CC_{1/2}$ | 0.998 (0.515) | 0.998 (0.447) | 0.995 (0.522) | 0.996 (0.500) | 0.996 (0.408) | 0.997 (0.510) | 0.998 (0.563) |
| $CC^*$ | 1.00 (0.825) | 0.999 (0.786) | 0.999 (0.828) | 0.999 (0.816) | 0.999 (0.761) | 0.999 (0.822) | 1.00 (0.849) |
| $I/\sigma I$ | 17.1 (1.4) | 15.1 (1.4) | 15.2 (1.4) | 15.2 (1.0) | 14.4 (1.3) | 17.5 (1.6) | 19.8 (1.4) |
| Completeness (%) | 99.7 (97.4) | 98.2 (96.9) | 99.6 (99.7) | 97.9 (89.6) | 99.2 (93.9) | 100 (100) | 99.9 (100) |
| Redundancy | 7.3 (6.5) | 6.4 (6.2) | 5.3 (5.4) | 6.9 (6.1) | 5.9 (3.4) | 7.5 (6.4) | 6.9 (7.0) |
| **Refinement** | | | | | | | |
| Resolution (Å) | 36.0–2.05 | 42–2.05 | 41.4–2.20 | 33.6–1.55 | 34.3–1.75 | 41.3–1.85 | 38.9–1.85 |
| No. reflections | 53178 | 50592 | 43011 | 120366 | 86108 | 71615 | 72890 |
| $R_{work}/R_{free}$ | 0.170/0.201 | 0.170/0.203 | 0.170/0.206 | 0.130/0.165 | 0.160/0.179 | 0.155/0.184 | 0.162/0.194 |
| *Non-H atoms:* | | | | | | | |
| Protein/DNA | 5819 | 5790 | 5796 | 5891 | 5907 | 5885 | 5858 |
| Ligand/ion | 35 | 24 | 34 | 30 | 29 | 45 | 41 |
| Water | 563 | 665 | 387 | 1115 | 975 | 735 | 466 |
| *B-factors (Å$^2$):* | | | | | | | |
| Protein/DNA | 41.5 | 37.2 | 49.5 | 26.5 | 32.0 | 39.6 | 43.6 |
| Ligand/ion | 36.8 | 25.8 | 45.6 | 27.1 | 32.2 | 43.9 | 59.1 |
| Water | 44.4 | 43.0 | 48.5 | 44.1 | 43.6 | 47.1 | 48.3 |
| *R.m.s. deviations:* | | | | | | | |
| Bond lengths (Å) | 0.006 | 0.004 | 0.005 | 0.006 | 0.004 | 0.006 | 0.004 |
| Bond angles (°) | 0.81 | 0.71 | 0.73 | 0.86 | 0.706 | 0.86 | 0.73 |

Each data set was collected from a single crystal. Values in parentheses are for highest-resolution shell (10% of reflections)

5′-phosphate. Together with K746, these lysines are suitably positioned to stabilize the AMP leaving group during nick sealing.

Notably, crystals produced in 2 mM Mg$^{2+}$ showed only a single tightly coordinated Mg$^{2+}$ at Site 1, suggesting tight binding for this magnesium ion. Given that Site 4 ligands are not conserved, we focused our attention on the functional role of Site 1. The octahedral coordination of the partially hydrated Mg$^{HiFi}$ involves interactions with the phosphodiester backbone between the −3 and −4 nucleotides (relative to the 3′-OH of the nick) as well as two glutamate residues from the protein, Glu 346 in the DBD and Glu 592 in the AdD (Fig. 2a, d). Both residues are stringently conserved in LIG1 homologs, including budding yeast Cdc9p and fission yeast cdc17 (Fig. 2c). The backbone carbonyl oxygens of L345 and P341 also participate in hydrogen bonding to Mg$^{HiFi}$ water ligands (Fig. 2d). Structurally, Mg$^{HiFi}$ mediates key interactions with the 3′-hydroxyl (3′-OH) strand, appearing to restrict the conformation of the strand in the ligase active site. The metal-reinforced DNA-binding mode secures the 3′-end in an A-form-like geometry, with the three terminal nucleotides exhibiting C3′-endo sugar puckers (Fig. 2a). The conspicuous location at the junction between the AdD and the DBD and the direct interaction with the DNA highlights a putative role of the Mg$^{HiFi}$ site in defining the conformation of the protein around the DNA substrate and proper positioning of the 3′-OH strand of a DNA nick within the active site.

**Mg$^{HiFi}$ metal site confers intrinsic LIG1 catalytic fidelity.** Given that Mg$^{HiFi}$ lies at the nexus of the LIG1–DNA interface, we hypothesized that the cation may control the ability of LIG1 to discriminate against nicks bearing damaged bases. To define the functional contribution of Mg$^{HiFi}$, we purified a double alanine mutant of LIG1 (LIG1$^{E346A/E592A}$) that removes both of the protein ligands (Supplementary Fig. 1A). Using a gel-based ligation assay, the concentration of active enzyme was determined and the extent of adenylylation of the purified enzyme was comparable to that of the LIG1$^{WT}$ (Supplementary Fig. 1B, C). We then performed a comprehensive survey of steady-state kinetic experiments with LIG1$^{E346A/E592A}$ and compared the results with those of the WT enzyme to discern any changes in the biochemical behavior upon removal of the Mg$^{HiFi}$ ligands (Supplementary Fig. 2). LIG1$^{E346A/E592A}$ exhibited a two-to-threefold reduction in k$_{cat}$ and approximately twofold reduction in the $K_M$ value for ATP relative to WT LIG1 (Supplementary Fig. 2 and Supplementary Table 2). The Mg$^{2+}$-dependence of the multiple turnover ligation reaction at saturating ATP and DNA yielded identical values for Mg$^{2+}$ activation ($K_{Mg}$) with LIG1$^{WT}$ and LIG1$^{E346A/E592A}$ (Supplementary Fig. 3A; Supplementary Table 3). We repeated the steady-state DNA dependence at a more physiologically relevant Mg$^{2+}$ concentration (1 mM free) and found that the K$_M$ value for the DNA substrate is improved by 24-fold for the LIG1$^{E346A/E592A}$ relative to LIG1$^{WT}$, reflecting a ~10-fold improvement in catalytic efficiency for LIG1$^{E346A/E592A}$ (Supplementary Fig. 4A and Supplementary Table 4). Collectively these results suggest that the Mg$^{HiFi}$ ion has an important role in tuning the interaction of LIG1 with the DNA substrate.

Previous work showed that 8oxoG can be incorporated during 1-nt gap filling base excision repair reactions and that LIG1

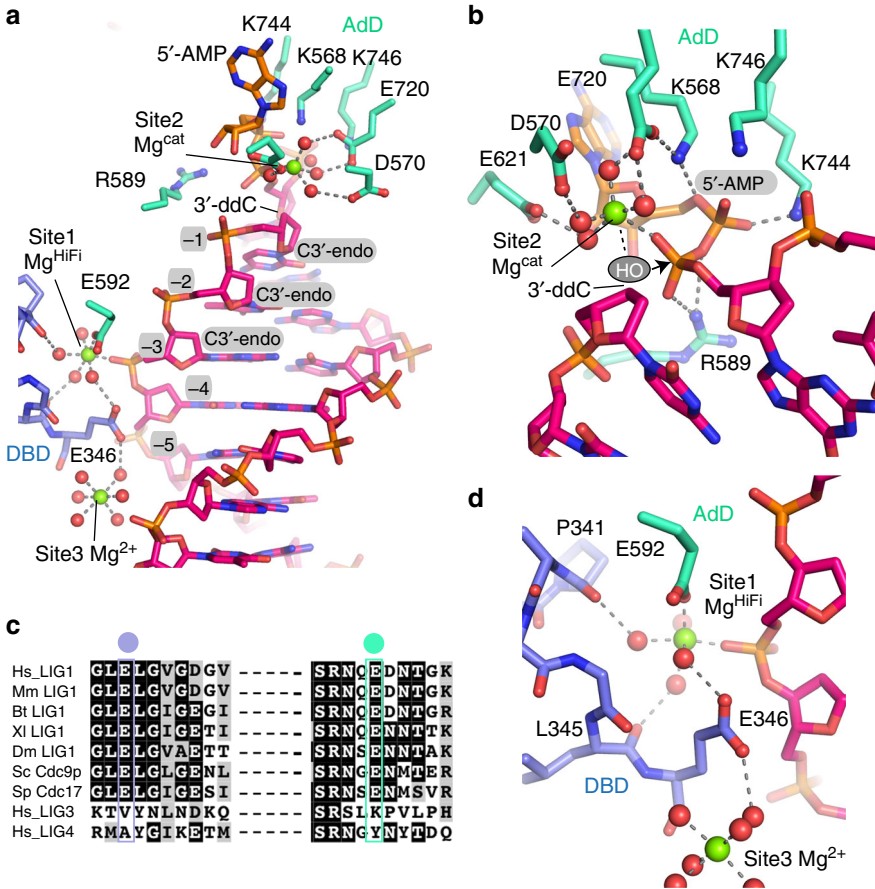

**Fig. 2** Active site and Mg$^{HiFi}$ metal-binding site geometry and conservation. **a** Divalent metal-binding sites in the LIG1•DNA complex. **b** Configuration of the LIG1 active site poised for nick sealing. **c** Sequence alignment of LIG1 homologs and human ligases generated by Clustal Omega shows HiFi site residues E346 and E592 are stringently conversed among LIGI homologs, but not in LIG3 and LIG4. **d** The high-fidelity metal site of LIG1 is scaffolded by glutamate side chains from the AdD (E592, cyan) and the DBD (E346, blue).

discriminates against 8oxo-2'-deoxyG (8oxoG) base pairs 8oxoG: C and 8oxoG:A at this 3' position[18–20]. Thus, we tested whether the Mg$^{HiFi}$ cation has a role in substrate discrimination by evaluating the ability of LIG1$^{E346A/E592A}$ to ligate substrates containing 8oxoG at the 3'-terminal position of a 28-mer substrate. Consistent with previous reports, we find LIG1$^{WT}$ is greatly compromised in its ability to complete ligation of the 8oxoG:C substrate, instead aborting ligation and leading to an accumulation of the AMP–DNA intermediate (Fig. 3a–c). Although LIG1$^{WT}$ can catalyze ligation of the 8oxoG:A substrate, it does so slowly, with a significant (~70%) accumulation of the adenylylated DNA intermediate. In contrast, the LIG1$^{E346A/E592A}$ mutant enzyme has dramatically enhanced activity on the 8oxoG-containing substrates. This mutant fully ligates the 8oxoG:A substrate and ligates approximately half of the 8oxoG:C substrate that it engages (Fig. 3c). Congruent with this decrease in abortive ligation, the LIG1$^{E346A/E592A}$ mutation substantially increases the catalytic efficiency for ligation of both 8oxoG-containing substrates (Fig. 3b and Supplementary Table 4). These observations demonstrate that Mg$^{HiFi}$ contributes to the discrimination against damaged base pairs immediately upstream of the nick, raising the question of how disruption of the HiFi site increases the catalytic efficiency of LIG1.

We next characterized the single-turnover ligation kinetics of the LIG1$^{E346A/E592A}$ mutant using a rapid chemical quench approach[30]. Following DNA substrate engagement, DNA ligase

catalyzes two chemical steps to join DNA ends together—adenylyl transfer and nick sealing (Fig. 1a). Rate constants for both steps were determined by monitoring a time course of the formation of ligated product and the build-up and consumption of the adenylylated DNA intermediate (Fig. 3d). LIG1$^{WT}$ and LIG1$^{E346A/E592A}$ were purified as the adenylylated enzyme and therefore no ATP was required for these reactions to occur. By performing rapid-quenching experiments over a range of Mg$^{2+}$ concentrations with the 28-mer-C:G substrate, we obtained the maximal rates of both adenylyl transfer (k$_{transfer}$) and nick sealing (k$_{seal}$). Both enzymes exhibited similar maximal rates for both chemical steps (Supplementary Fig. 3 and Supplementary Table 4). These data demonstrate that the LIG1$^{E346A/E592A}$ mutant does not perturb catalysis with a normal DNA substrate and suggest that the lower k$_{cat}$ value for LIG1$^{E346A/E592A}$ is owing to a step prior to adenylyl transfer or following nick sealing.

The transient kinetic experiments were extended to investigate the microscopic rates of ligation for both 8oxoG substrates at a physiological concentration of Mg$^{2+}$ (Fig. 3d). These data reveal that LIG1$^{WT}$ discriminates strongly against 8oxoG:C at both the adenylyl transfer and nick-sealing steps (Fig. 3e, f). Discrimination against 8oxoG:A also occurs in both steps, but to a lesser extent than with the 8oxoG:C substrate, especially in the adenylyl transfer step. Unexpectedly, the kinetics of adenylyl transfer and nick sealing for ligation of the 8oxoG substrates are not significantly altered for LIG1$^{E346A/E592A}$ (Fig. 3e, f).

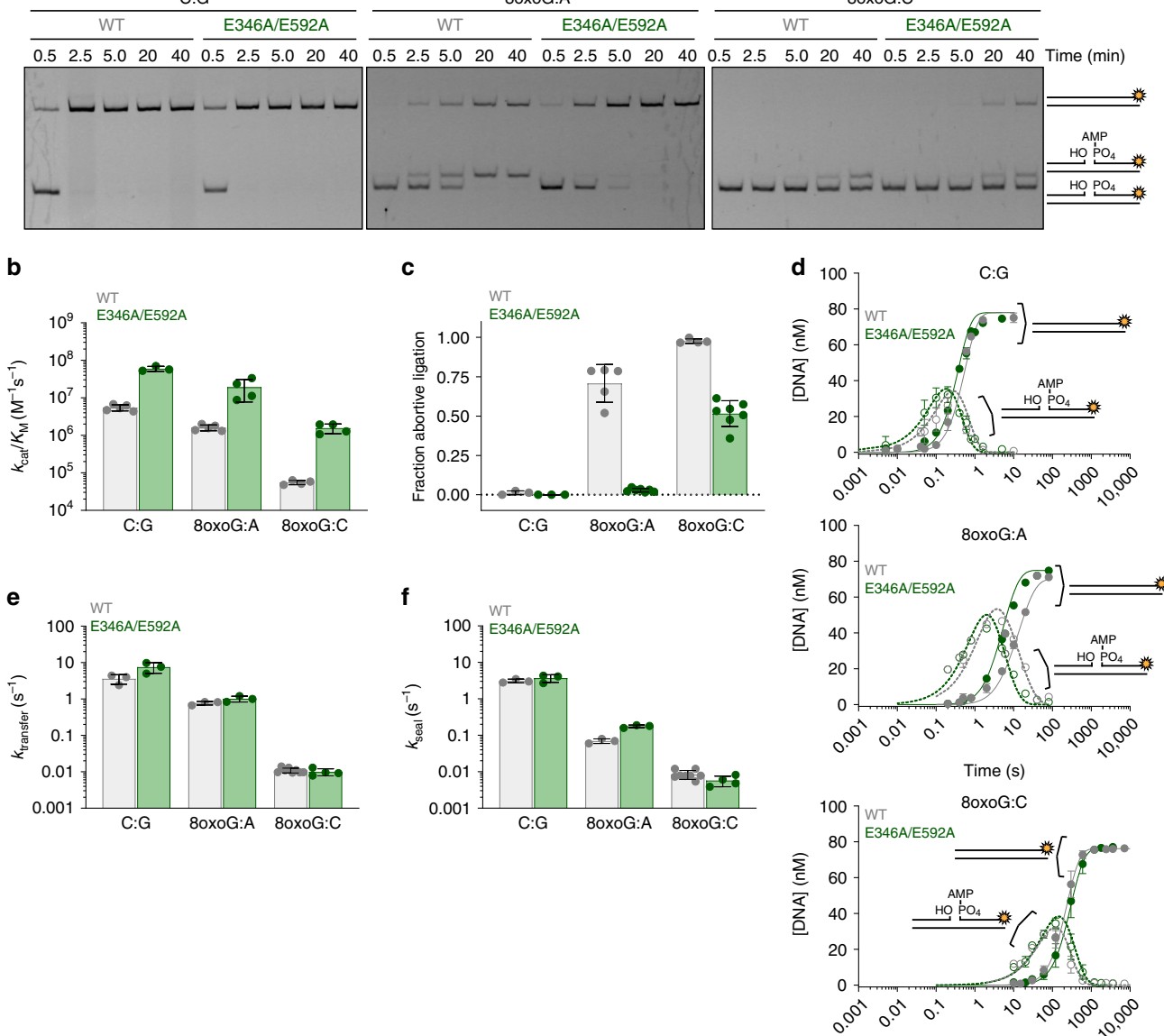

**Fig. 3** Impact of the Mg$^{HiFi}$ metal-binding site on LIG1 activity and fidelity. **a** Representative denaturing gel image of steady-state ligation reactions in the presence of 2 mM MgCl$_2$ and 1 mM ATP. Reactions contained 5 nM of Δ232 LIG1$^{WT}$ or LIG1$^{E346A/E592A}$ and 500 nM DNA substrate with a C:G, 8oxoG:C or 8oxoG:A pair at the −1 nt position. **b** Relative $k_{cat}/K_M$ values for ligation of the various DNA substrates determined by steady-state ligation kinetics (Supplementary Fig. 4). **c** The fraction of abortive ligation events determined from steady-state ligation reactions containing 1 or 10 nM LIG1, 2 μM DNA substrate, 1 mM ATP and 2 mM MgCl$_2$ (1 mM Mg$^{2+}$$_{free}$). **d** Representative pre-steady-state ligation by LIG1$^{WT}$ or LIG1$^{E346A/E592A}$ in the presence of 1 mM MgCl$_2$ measured with a rapid quench apparatus. Product formation (closed symbols, solid line) and intermediate formation and decay (open symbols, dashed line) are shown with their respective fits. Rate constants for adenylyl transfer **e** and nick sealing **f** were determined from Berkeley–Madonna fits. All data are the mean ± S.D. of ≥3 replicates. Source data for panels are provided as a Source Data file.

**Constrained 3′-strand binding is scaffolded by the HiFi Mg$^{2+}$.**
Our data show that the LIG1$^{E346A/E592A}$ mutation confers increased catalytic efficiency on damaged and undamaged DNA substrates (Fig. 3b). As ablation of the Mg$^{HiFi}$ site had significant impacts on substrate interactions, we examined the structural consequences of this mutation (Supplementary Fig. 5). Crystallographic isomorphism between the LIG1$^{WT}$•DNA crystal structures enabled direct assessment of conformational changes in model-phased Fo (Mutant)-Fo (WT) electron density maps (Supplementary Fig. 5A). Removal of the Mg$^{HiFi}$ ligands led to the loss of bound metal ions at Sites 1 and 2 and concerted structural rearrangements of the DNA in the upstream (3′-OH

side) of the bound nicked duplex (Supplementary Fig. 5B). These rearrangements included 0.4–1.6 Å shifts of the upstream duplex, whereas the protein and downstream duplex remained largely static (Supplementary Fig. 5A, B). The most significant motions are localized to 3′-terminal base pairs, -1 through -4, relative to the 3′-OH. Removal of Mg$^{HiFi}$ causes rearrangements in protein–DNA van der Waals contacts proximal to the mutation, creating a cavity that allows for alternative conformations of the DNA backbone and dynamic binding of the phosphodiester backbone between the -1 and -2 nucleotides (Supplementary Fig. 5C, D). Overall, mutation of the Mg$^{HiFi}$ site has localized impacts on the trajectory of the upstream strand.

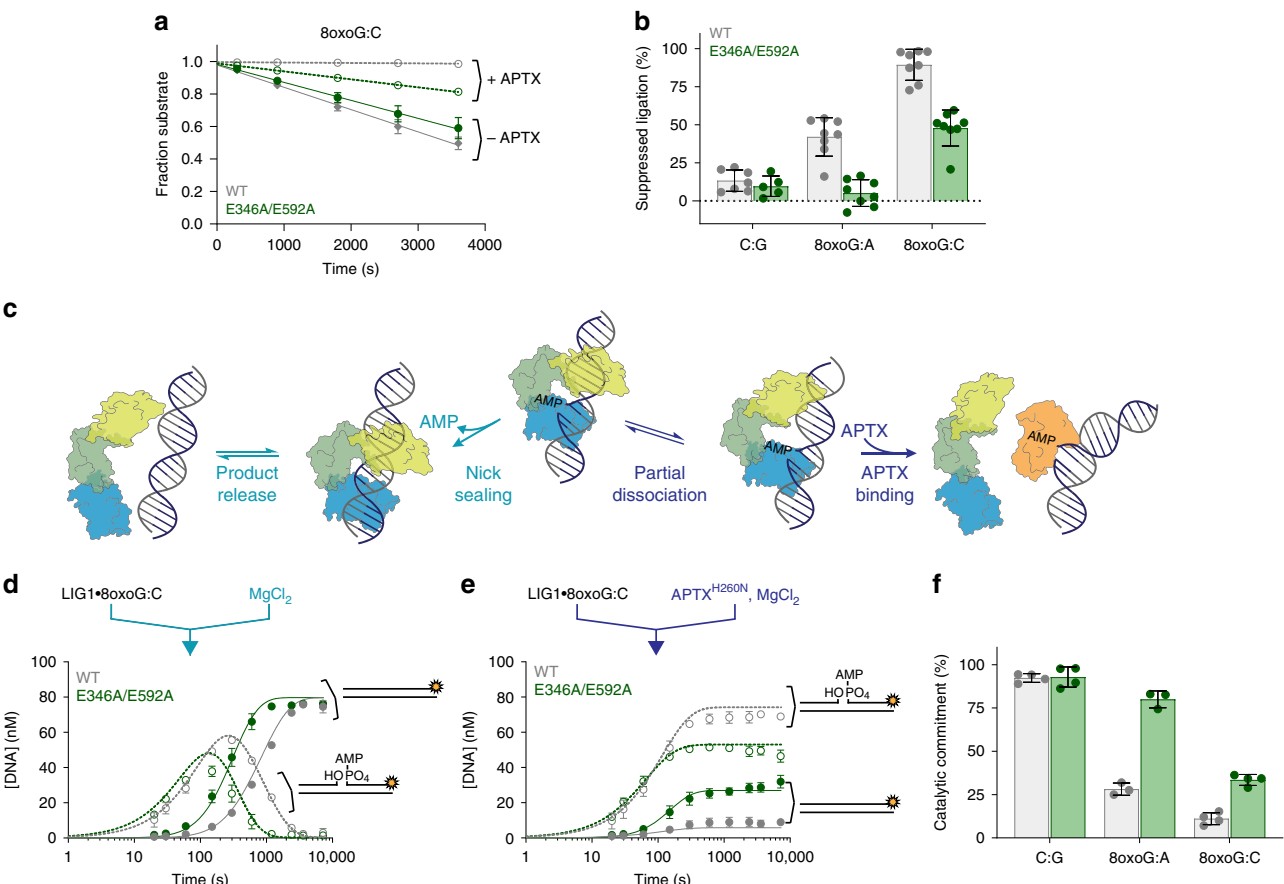

**Fig. 4** APTX suppresses LIG1-catalyzed ligation of 8oxoG-containing DNA. **a** Representative time courses showing the consumption of substrate in steady-state ligation reactions containing 10 nM LIG1[WT] or LIG1[E346A/E592A], 500 nM 8oxoG:C DNA substrate, 1 mM ATP and 2 mM MgCl$_2$ (1 mM Mg$^{2+}$$_{free}$). Reactions were performed in the absence (solid symbols, solid lines) or presence (open symbols, dashed lines) of 1 nM APTX[WT]. The percent change in initial rate from reactions in the absence of APTX[WT] to those in the presence of APTX is plotted in **b** to illustrate the extent to which APTX[WT] suppresses LIG1 activity. Time courses for reactions with the C:G and 8oxoG:A substrates are shown in Supplementary Fig. 6. **c** Kinetic model for trapping of the AMP–DNA intermediate by the APTX[H260N] catalytically inactive mutant. Partial dissociation of LIG1 gives the opportunity for APTX to capture the intermediate. Pre-steady-state reactions containing 800 nM LIG1, 80 nM 8oxoG:C, 1 mM Mg$^{2+}$$_{free}$, and 1 mM ATP were performed in the absence **d** or presence **e** of APTX[H260N]. Product formation (closed symbols) and intermediate formation (open symbols) are shown with their respective fits determined using Berkeley–Madonna (See Supplementary Fig. 9 for the model). **f** Catalytic commitment was determined as the proportioning between nick sealing and dissociation from the AMP–DNA intermediate. These data demonstrate that LIG1[E346A/E592A], which disrupts the Mg[HiFi] site, has increased commitment to nick-sealing of an 8oxoG:C substrate relative to LIG[WT]. Data for the catalytic commitment to ligation of the G:C and 8oxoG:A substrates are shown in Supplementary Fig. 7. All data are reported as the mean ± S.D. of ≥3 replicates. Source data for panels are provided as a Source Data file.

Paradoxically, disruption of the Mg[HiFi] site has marked impacts on the DNA substrate K$_M$ values (24-fold for undamaged C:G, 51-fold for 8oxoG:C, and 15-fold for 8oxoG:A; Supplementary Table 4). Thus, the combined structural and kinetic data together indicate that Mg[HiFi] is dispensable for the chemical steps, but likely influences other steps that precede or follow the adenylyl transfer and nick-sealing steps, such as substrate binding, enzyme/substrate conformational rearrangements, and/or product release.

**APTX suppresses LIG1 reactions on 8oxoG-damaged substrates.** The biochemical characterization of LIG1 3′-end recognition demonstrated the importance of abortive ligation as a bypass mechanism to limit the sealing of damaged termini. If adenylylation of an improper nick occurs, then the Mg[HiFi] site architecture leads to enhanced partitioning away from erroneous ligation and towards abortive ligation. Abortive ligation on damaged nicks could provide additional time for other DNA

repair pathways to operate. However, the 5′-AMP moiety must be removed to restore a 5′-phosphate for ligation after 3′-proof-reading and gap filling. As APTX hydrolyses adenylylated 5′-termini, we hypothesized that the interplay between LIG1 and APTX would be especially important for controlling the pathway choice between erroneous ligation and abortive ligation. Therefore, we performed steady-state ligation assays in the presence or absence of human APTX.

Indeed, APTX is able to fully suppress abortive ligation by LIG1[WT] with the 8oxoG:C substrate at a 1:10 ratio of APTX: LIG1 (Fig. 4a, b). In contrast, LIG1[E346A/E592A] is only partially suppressed by APTX under these conditions. A similar trend was observed for the 8oxoG:A substrate, with APTX effectively suppressing the reaction catalyzed by LIG1[WT], whereas not affecting the reaction catalyzed by the LIG1[E346A/E592A] mutant (Fig. 4b and Supplementary Fig. 6B). These data suggest that occupancy of the Mg[HiFi] site produces strain in the LIG1•DNA complex, leading to increased abortive ligation with damaged termini and subsequent processing by APTX. This fidelity

comes at a cost to overall ligation efficiency, as APTX causes modest, but readily detectable suppression of ligation by LIG1$^{WT}$ with the undamaged substrate (Fig. 4b and Supplementary Fig. 6A). In all cases, disruption of the Mg$^{HiFi}$ site leads to increased catalytic efficiency and decreased overall ligation fidelity.

For APTX to engage with the adenylylated intermediate, it must be given access to the intermediate by LIG1, presumably by partial or complete undocking of LIG1 from the adenylylated intermediate (Fig. 4c). In ligation reactions that included ATP, we observed an extended lifetime for the adenylylated DNA intermediate in single-turnover experiments with the 8oxoG-containing substrates (Fig. 4d). This observation is consistent with transient binding of ATP to certain LIG1 conformations that selectively limits the population of LIG1 competent for nick sealing. Intriguingly, the delay in nick sealing is prolonged in LIG1$^{WT}$-catalyzed reactions compared with those containing LIG1$^{E346A/E592A}$ (Fig. 4d; Supplementary Fig. 7A, B). We posited that this delay may stem from sampling accessible LIG1 conformations that afford APTX the opportunity to sequester the adenylylated intermediate. To test the accessibility of the adenylylated intermediate to APTX, we made use of an APTX$^{H260N}$ mutant, which is catalytically inactive (Supplementary Fig. 8) but retains the ability to bind adenylylated ligation intermediates[24]. APTX$^{H260N}$ was included in single-turnover ligation experiments containing physiological concentrations of both Mg$^{2+}$ and ATP to assess the ability of both LIG1$^{WT}$ and LIG1$^{E346A/E592A}$ to remain committed to catalysis on the various substrates. In the presence of APTX$^{H260N}$, any adenylylated intermediate that is accessible should be sequestered by APTX$^{H260N}$ and LIG1 re-association will be blocked (Fig. 4c). In control reactions with the 28-mer-C:G substrate, both LIG1$^{WT}$ and LIG1$^{EE/AA}$ remained highly committed to ligating the substrate (Fig. 4f and Supplementary Fig. 7). With either of the two 8oxoG-containing substrates, however, a significant portion of the adenylylated intermediate was trapped by APTX$^{H260N}$ (Fig. 4d–f and Supplementary Fig. 7C, D). For LIG1$^{WT}$, there was a dramatic impact of APTX, with only 30% of ligation events with the 8oxoG:A substrate and about 10% of ligation events with the 8oxoG:C going to completion (Fig. 4e and Supplementary Fig. 7D). Thus, in the presence of APTX, the catalytic commitment of LIG1$^{WT}$ to erroneous ligation is reduced by 70 and 90%, respectively, for the 8oxoG:A and 8oxoG:C substrates (Fig. 4f). In contrast, LIG1$^{E346A/E592A}$ remained ~ 80% committed to catalysis with the 8oxoG:A substrate and ~30% committed with the 8oxoG:C substrate (Fig. 4f). Along with structural data detailing flexibility in the upstream strand when substrate is bound by LIG1$^{E346A/E592A}$, the ability of LIG1$^{E346A/E592A}$ to protect ligation intermediates from APTX sequestration implies that the Mg$^{HiFi}$ helps enforce a rigid upstream DNA structure. When a damaged nick is encountered, we hypothesize LIG1 adopts a more open conformation, allowing APTX to intercept the adenylylated intermediate.

**Molecular basis of mutagenic ligation by LIG1$^{E346A/E592A}$.** LIG1$^{E346A/E592A}$ confers a dramatically increased catalytic efficiency on oxidatively damaged substrates, but retains high selectivity in the chemical steps of catalysis, with a ~ 640-fold (8oxoG:C) and ~ 21-fold (8oxoG:A) slower $k_{seal}$ compared with an undamaged C:G base pair (Supplementary Table 4). These observations suggest that, in addition to Mg$^{HiFi}$-dependent substrate discrimination, alterations to the active site architecture also contribute to the intrinsic selectivity against ligation of damaged 3′ base pairs by LIG1. To evaluate the structural impacts of DNA damage on the LIG1 active site, we performed crystallization experiments on a number of combinations of LIG1 with 8oxoG-containing DNA substrates. Notably, LIG1$^{WT}$ complexes with both 8oxoG-containing DNA substrates and LIG1$^{E346A/E592A}$ bound to 8oxoG:C were all refractory to crystallization. However, we successfully crystallized and determined the structure of LIG1$^{E346A/E592A}$ in complex with 3′-8oxoG:A to 1.85 Å, as well as with an undamaged C:G base pair under similar conditions. In these structures, the native 3′-OH is present in the damaged and undamaged 3′-strands, and metal was chelated with ethylenediaminetetraacetic acid (EDTA) to prevent nick sealing. Despite inclusion of EDTA, we found that the 5′-termini are fully adenylylated in the crystals and poised for step 3 (nick sealing; Fig. 5).

The 8oxoG:A 3′-terminal pair adopts Hoogsteen base pairing geometry in the LIG1 active site, reminiscent of DNA polymerases that accommodate 8oxoG:A base pairs[31–33]. The *syn* geometry of the 8oxoG is stabilized by hydrogen bonding between the exocyclic 2-amino group and the phosphodiester backbone (Fig. 5a). In the undamaged complex, R589 makes bifurcated salt-bridging interactions with the 5′-5′ phosphoanhydride. An ordered hydration shell of tightly bound water molecules (Wat1 and Wat2, Fig. 5c) are putative proton donors to the leaving group AMP during catalysis. In contrast, significant rearrangements to catalytic architecture are observed near the damaged 8oxoG nucleotide. Flipping of 8oxoG into the major groove coincides with three key structural alterations (Figures 5B–D): disruption of the local hydration structure, concerted rearrangements of active site residues N590 and R589, and movement of the pyrophosphate towards R589. These observations suggest that the 8oxoG-damaged base pairing imparts non-native active site conformations that further suppress rates of the chemical steps of catalysis on the 8oxoG substrates.

**Metal binding enforces high fidelity DNA ligation.** Our X-ray structures of the E346 A/E592A mutant reveal a cavity that facilitates dynamic protein–DNA binding in the mutant ligase structure. We hypothesize that the impaired fidelity of the LIG1$^{E346A/E592A}$ enzyme results from relaxation of the strict geometric requirements for Mg$^{2+}$ metal coordination during DNA binding, and creation of this cavity. To more directly dissect the contribution of metal coordination to ligation fidelity, we generated an additional E592R mutation and assessed ligation and abortive ligation profiles on undamaged and 8oxoG-containing substrates. Similar to the LIG1$^{E346A/E592A}$ mutant, LIG1$^{E592R}$ displays markedly attenuated abortive ligation (Fig. 6a), and enhanced nick-sealing activity (Supplementary Fig. 10) on 8oxoG:A and 8oxoG:C substrates. We further determined the crystal structure of LIG1$^{E592R}$ at 1.85 Å (Fig. 6b, Table 1). In this structure, the R592 substitution occupies the approximate position of E592 plus the HiFi magnesium ligand, and forms a salt bridge to the phosphodiester backbone in place of the metal-DNA contact. Compared with LIG$^{WT}$ a small cavity is formed, but the interface is typified by a high complementary of the protein and 3′-DNA terminus. This cavity is reduced relative to that which is present in LIG1$^{E346A/E592A}$ (Fig. 6c, but very similar effects were observed on the fidelity of DNA ligation (Fig. 6a). Thus, substitution of a direct protein–DNA interface for the native protein–metal–DNA interface is not sufficient for high-fidelity ligation. We conclude that high fidelity of LIG1 is dictated by the extended DNA-binding interactions that converge on the nucleic acid backbone at the nexus of the 3′-strand. Metal coordination imparts strict geometric requirements between the DBD, AdD, and DNA and serves as a precise sensor of base pairing interactions in the 3′-strand.

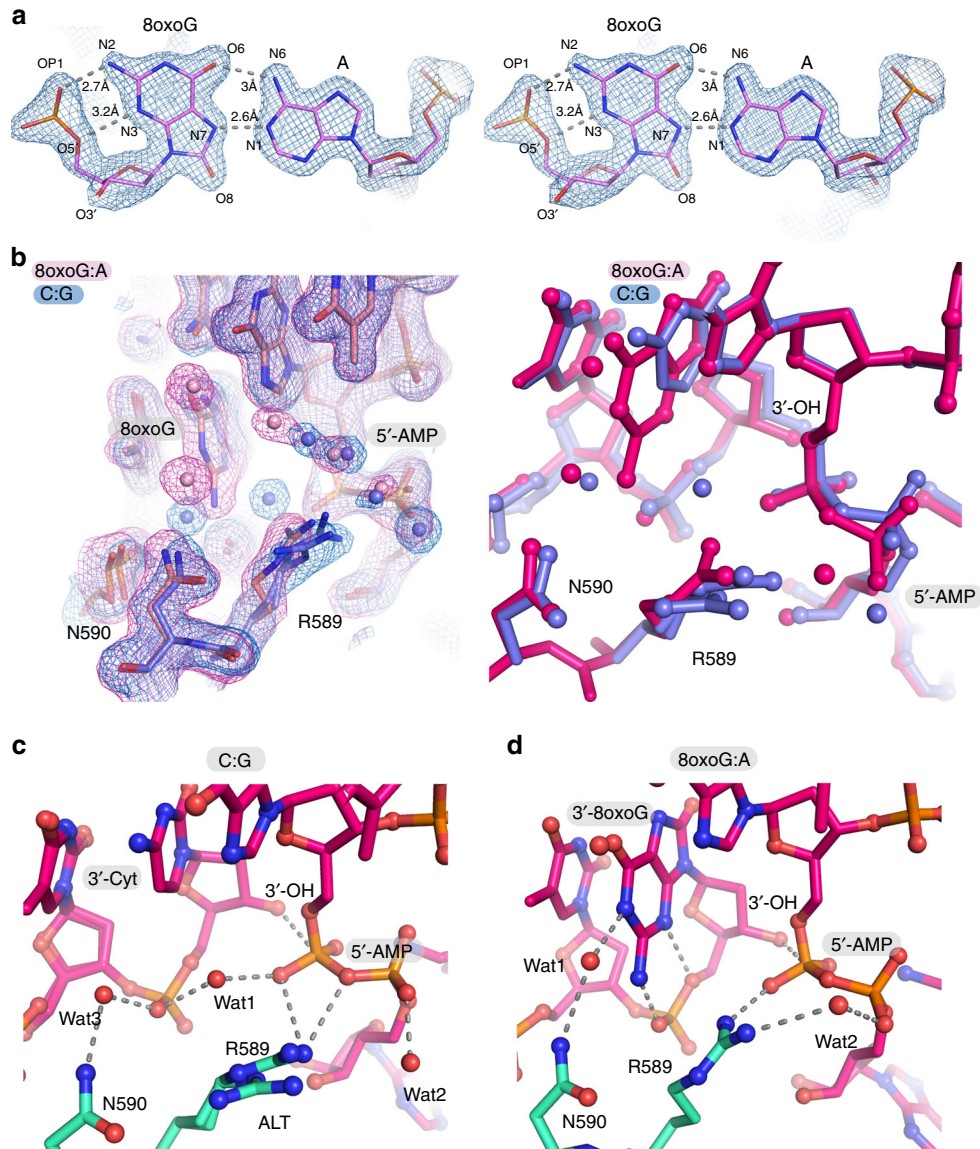

**Fig. 5** LIG1[E346A/E592A] structures in complex with damaged 8oxoG:A DNA. **a** Stereo view of final model-phased 2Fo-Fc electron density contoured at 2.0 σ and displayed for the 8oxoG:A nucleotide pair bound in the E346A/E592A double mutant structure shows the pairing is in a Hoogsteen configuration (A anti:8oxoG syn). The N2 and N3 atoms of the 8oxoG make additional contacts with the phosphate backbone. **b** Left: superposition of the refined 2Fo-Fc electron density for the C•G (blue) and 8oxoG•A (red) bound LIG1[E346A/E592A] mutant complexes. Right: stick model superposition. **c** Active site architecture of the undamaged complex. **d** Active site architecture of the 8oxoG:A complex.

## Discussion

The fidelity of DNA ligation has long been an underappreciated component of the faithful replication and repair of genomic DNA. In this work, we used structural and quantitative biochemical approaches to elucidate the mechanisms by which the replicative human DNA ligase, LIG1, discriminates against improper 3′ termini (Fig. 7a). Our study reveals that LIG1 discriminates against damaged 3′ termini at both steps 2 (adenylyl transfer) and 3 (nick sealing) of the ligase reaction. This intrinsic fidelity hardwires LIG1 to prevent deleterious abortive and mutagenic ligations. We further establish that the Mg[HiFi] site plays an integral role in enforcing recognition of the 3′-strand and enables an additional layer of surveillance to bias improper nicks toward abortive ligation. APTX is able to access the stalled AMP–DNA intermediate leading to dissociation of LIG1 and restoration of the 5′-phosphate terminus (Fig. 7b). In this role, APTX efficiently suppresses abortive and mutagenic ligation events. Disabling the high-fidelity mechanism via removal of the Mg[HiFi] site significantly reduces the ability of the enzyme to discriminate against improper nicks, nonspecifically increases ligation efficiency on both undamaged and damaged nicks, and permits the enzyme to evade APTX surveillance (Fig. 7b).

The cells from AOA1 patients who suffer from deficiency in APTX have previously been found to be sensitized to agents that cause oxidative damage and this results in elevated genomic instability[12,34]. The observation herein that APTX enhances fidelity of LIG1 supports a role of APTX in the response to oxidative DNA damage and it is possible that reduced fidelity of LIG1-mediated DNA repair reactions plays a role in the etiology of this disease. Although this hypothesis is yet to be tested, the physiological roles of APTX are considered to be complex as it also interacts with several proteins related to the DNA damage response, including XRCC1 and XRCC4[35].

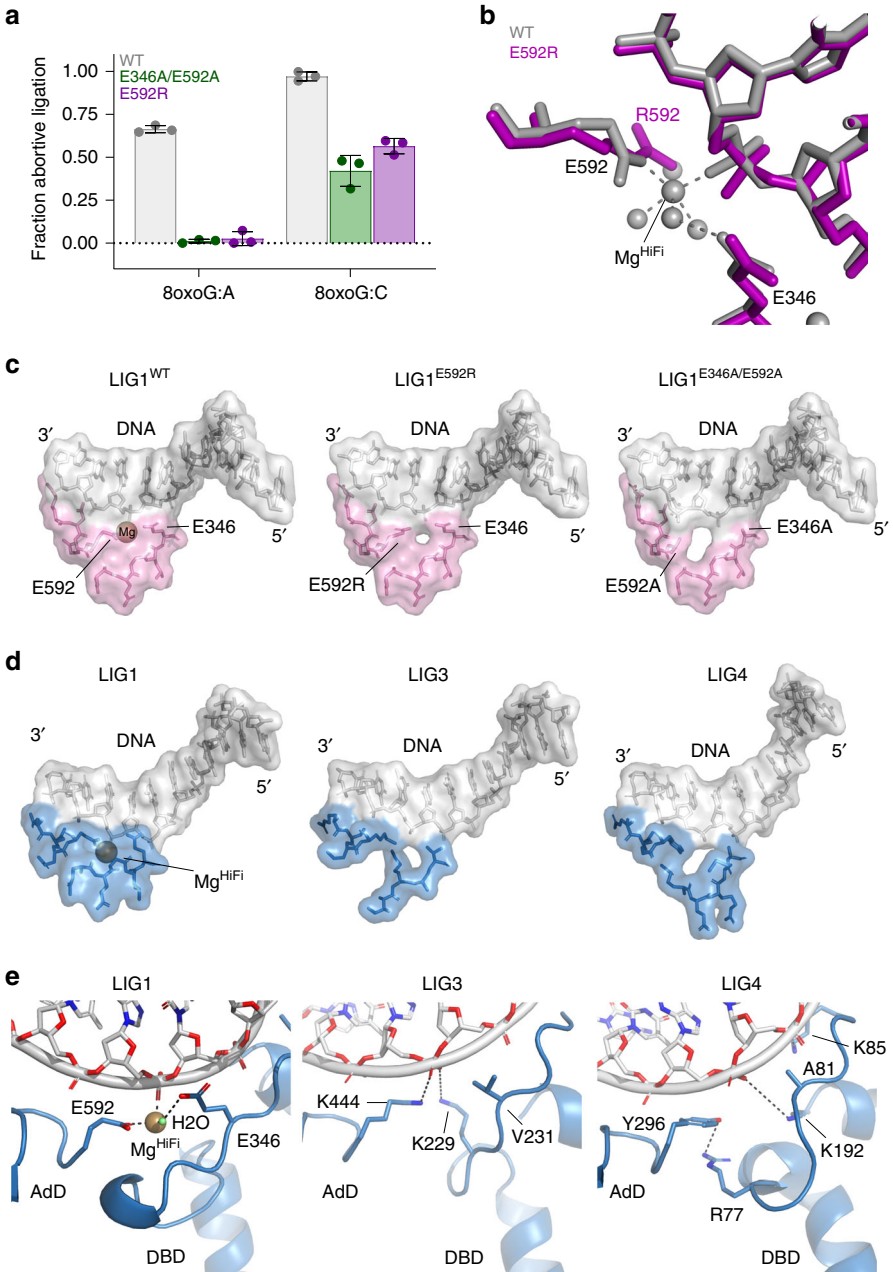

**Fig. 6** The HiFi protein–metal–DNA interface is required for high-fidelity ligation. **a** Analysis of abortive ligation by WT and mutant LIG1. The fraction of abortive ligation events determined from steady-state ligation reactions containing 1 or 10 nM LIG1, 2 μM DNA substrate, 1 mM ATP, and 2 mM MgCl$_2$ (1 mM Mg$^{2+}_{\text{free}}$). **b** X-ray structure of LIG1$^{E592R}$ (purple) overlaid upon LIG$^{WT}$ (grey). **c** Comparison of WT and mutant LIG1 3′-strand binding. Surface representations of the DNA are shown in gray, with the protein in pink. **d** Comparison of LIG1 3′ strand binding to human LIG3 (PDB 3L2P) and LIG4 (PDB 6BKF). Surface representations are displayed with DNA in grey and protein in blue. **e** Cartoon display of LIG1 comparisons with LIG3 and LIG4. Source data for panels are provided as a Source Data file.

The Mg$^{HiFi}$ site is conserved in other eukaryotic replicative DNA ligases, but it is notably absent in the human LIG3 and LIG4 enzymes. A comparison of the 3′-strand binding properties of LIG1 with LIG3 and LIG4 is shown in Fig. 6d and 6E[7–9]. LIG1 exhibits close surface complementarity that extends from the nick through the N-4 position (Fig. 6d). Notably, although the protein–DNA contacts of LIG3 and LIG4 are mediated by topologically equivalent DNA-binding loops, these loops adopt variable structures in the three DNA ligases (Fig. 6d, e). The LIG3 and LIG4 protein–DNA interface are comparatively discontinuous and are interrupted with cavities, resembling the LIG1$^{E346A/E592A}$ and LIG1$^{E592R}$ low fidelity mutants. We speculate that discontinuous 3′-strand binding contributes to the relaxed fidelity for 3′ mismatches observed for human LIG3 compared to LIG1[17], and the mutagenic DNA repair reactions performed during DSB repair by LIG4[10,11]. It will be important to address to what extent the other human DNA ligases employ 3′-OH strand-binding interactions to control substrate specificity. Variations in this interface could explain differences in fidelity and efficiency of ligation among the different DNA ligases. Our

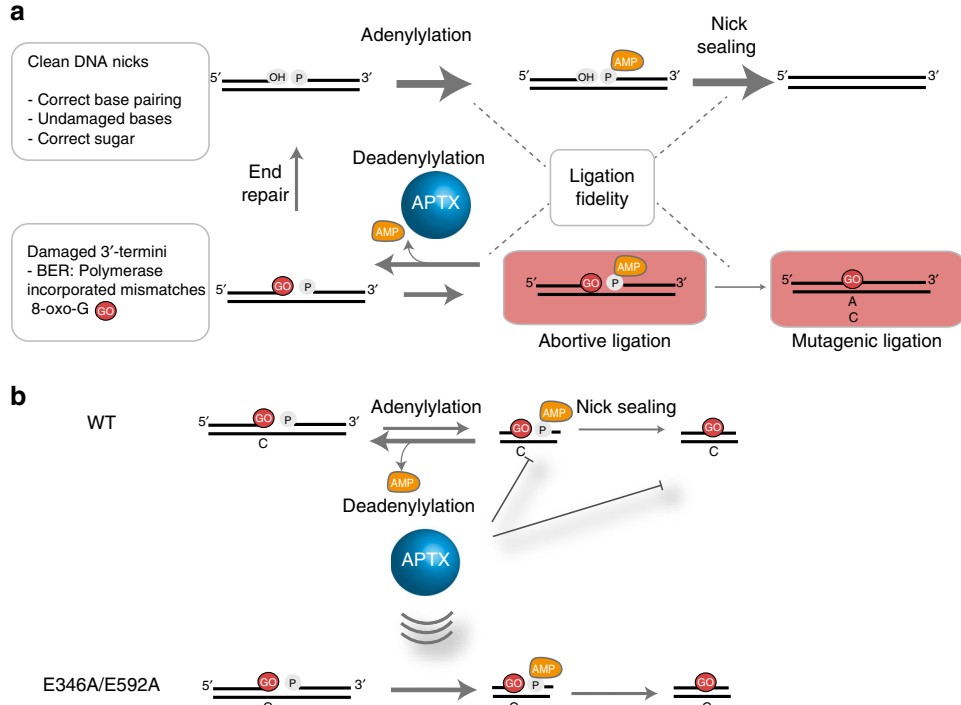

**Fig. 7** Summary of two-tiered fidelity of DNA ligation by LIG1 and APTX. **a** LIG1 discriminates against 3′ damage (exemplified by 8oxoG:C) at both the adenylylation and nick-sealing steps. The width of the arrows indicates the relative rates of the different steps. Abortive ligation forms adenylylated DNA that can be reversed by the hydrolytic action of APTX. This gives an opportunity for other forms of DNA repair, such as exonucleolytic proofreading. **b** APTX increases fidelity of LIG1 by capturing and hydrolyzing the AMP–DNA intermediate, preventing accumulation of abortive, and mutagenic ligation products. Mutations ablating the high-fidelity $Mg^{2+}$ site decreases abortive ligation and surveillance by APTX.

work indicates that the architecture of the $Mg^{HiFi}$ site in LIG1 is a distinguishing structural feature that is integral to the high-fidelity ligation activity of this replicative DNA ligase.

## Methods

**Protein expression and purification for crystallization**. LIG1[E346A/E592A] was generated using QuikChange site-directed mutagenesis (Stratagene). LIG1 proteins (aa262–904) were expressed in *E. coli* Rosetta 2 (DE3) cells. Cell cultures were grown at 37 °C in Terrific Broth supplemented with ampicillin (100 μg mL⁻¹) and chloramphenicol (34 μg mL⁻¹) until $A_{600}$ reached 1, at which point 50 μM isopropyl β-D-thiogalactopyranoside (IPTG) was added. Protein expression was carried out at 16 °C overnight. Cells were harvested by centrifugation (5000 rpm, 20 min). Cell pellets were suspended and lysed in 30 mL lysis buffer (50 mM Tris pH 8.5, 500 mM NaCl, 10 mM imidazole, 0.1 g/L lysozyme, 1 tablet Roche mini EDTA-free protease inhibitor cocktail) at 4 °C for 30 min, followed by sonication. The cell lysate was fractionated by centrifugation (13,000 rpm, 20 min, 4 °C). The soluble fraction was applied to Ni-NTA resins (5 mL packed volume) which has been equilibrated with 15 mL (50 mM Tris-Cl), pH 8.5, 500 mM NaCl, 10 mM imidazole). The column was washed with 100 mL (50 mM Tris-Cl, pH 8.5, 500 mM NaCl, 10 mM imidazole), 15 mL (50 mM Tris-Cl), pH 8.5, 500 mM NaCl, 30 mM imidazole), and the His-tagged protein was eluted in 15 mL (50 mM Tris-Cl, pH 7.5, 500 mM NaCl, 300 mM imidazole). The His-tag was removed by TEV protease at 4 °C overnight. The untagged protein was purified on HiLoad 16/600 Superdex 200 gel filtration column in buffer (25 mM Tris-Cl), pH 7.5, 150 mM NaCl, 1 mM TCEP, 0.1 mM EDTA), followed by HiTrap SP HP 5 mL cation exchange column (low salt buffer: 20 mM Tris-Cl), pH 7.5, 0.2 mM EDTA, 1 mM TCEP; high salt buffer: 20 mM Tris-Cl), pH 7.5, 1 M NaCl, 0.2 mM EDTA, 1 mM TCEP). The quality of the purified proteins was analyzed by sodium dodecyl sulfate polyacrylamide gel electrophoresis (SDS-PAGE). Freshly purified proteins were concentrated in Amicon concentrators (10 K MW cutoff) and used immediately in crystallization experiments.

**Protein expression and purification for kinetic analysis**. The plasmid for expression of the catalytic domain of human LIG1 (232–919) used for kinetic analysis is described elsewhere[7,8]. Mutant Δ232 LIG1 vectors were constructed using site-directed mutagenesis with synthetic primers and confirmed by sequencing of the coding region. The Δ232 LIG1 constructs were transformed and expressed in *E. coli* BL21(DE3) Rosetta 2 cells. Cells were grown in a shaker at 37 °C in Terrific Broth media supplemented with auto-induction components until they reached an $A_{600}$ nm of ~1. The shaker was then turned down to 16 °C and

expression proceeded for 16–18 h. Cells were harvested by centrifugation, resuspended in lysis buffer (50 mM Tris-Cl at pH 7.5, 10% glycerol, 300 mM NaCl, 5 mM BME and 0.1% IGEPAL CA-630 (Sigma-Aldrich)), and frozen at −80 °C.

After thawing, protease inhibitors (0.5 mM PMSF, 0.5 μg/mL leupeptin, 0.7 μg/mL pepstatin A) were added to cells. Cells were lysed by three rounds of passaging through an EmulsiFlex-C3 (Avestin) with 20 kpsi pressure. After centrifugation of the lysate, polyethyleneimine was added to the soluble extract to a final concentration of 0.1%. Following a second centrifugation step, soluble extract was bound to a HisTrap column (GE) equilibrated with buffer 1 (20 mM HEPES at pH 7.5, 0.1% IGEPAL CA-630, 500 mM NaCl, 20 mM imidazole). The column was washed with buffer 1, followed by buffer 2 (buffer 1 with 50 mM imidazole). The protein was eluted from the column using a 0–100% gradient of buffer 3 (20 mM HEPES at pH 7.5, 0.1% IGEPAL CA-630, 100 mM NaCl, 50 mM imidazole) to buffer 4 (buffer 3 with 300 mM imidazole). The His-tag was cleaved using PreScission protease, and a mixture of 1 mM ATP and 10 mM MgCl₂ was added to the pooled fractions to ensure adenylylation of Δ232 LIG1. Proteins were further purified with HiTrap Q (GE) and HiTrap Blue (GE) columns. Following purification, Δ232 LIG1[WT] and LIG1[E346A/E592A] proteins were dialyzed into storage buffer (25 mM Tris-Cl, pH 7.5, 150 mM NaCl, 1 mM DTT, and 0.1 mM EDTA and stored at −80 °C). Protein concentrations were estimated using absorbance at 280 nm. Active enzyme concentrations were determined using single-turnover ligation reactions, in which increasing concentrations of LIG1 were incubated with a fixed concentration of the fluorescein-labeled nicked 28-mer substrate[30].

**APTX expression and purification**. An *E. coli* codon-optimized coding sequence (GenScript) facilitates robust recombinant overexpression of human APTX[24]. WT and H260N human APTX catalytic domain (cat, residues 165–342) were expressed from pET15b as N-terminal 6× His-tagged proteins in E.coli BL21(DE3) codon-plus cells (Novagen). Cell cultures were grown at 37 °C in LB medium containing ampicillin (100 μg mL⁻¹) and chloramphenicol (34 μg mL⁻¹) until the $A_{600}$ reached 0.8–1, at which time cells were cooled to 16 °C, and grown for an additional 8–12 h, without IPTG induction. Cells were lysed by sonication in lysis buffer (50 mM Tris, pH 8.5, 500 mM NaCl, 10 mM imidazole, 0.01 g/L lysozyme, with Roche mini EDTA-free protease inhibitor). The soluble lysate was applied to Ni-NTA column (5 mL, Qiagen) and 6× His-tagged APTX proteins were eluted in lysis buffer with 300 mM imidazole. The 6× His-tag was removed with overnight thrombin digestion (50U) (Sigma) at 4 °C. Subsequent purification was achieved by size exclusion chromatography (Superdex 75, GE healthcare) in 50 mM Tris, pH 7.5, 500 mM NaCl, 5% glycerol, 0.1% β-mercaptoethanol) and cation exchange chromatography on a 5 mL Hi-Trap SP HP (GE Healthcare).

**Crystallization and structure determination**. All LIG1–DNA complex crystals were grown by hanging drop. 1 μL of protein•DNA complex solution (20 mg mL$^{-1}$ LIG1, 1.5:1 DNA:protein molar ratio, in 150 mM NaCl, 20 mM Tris-Cl, pH 7.5, 1 mM TCEP, and 1 mM ATP) with an equal volume of precipitant solution (100 mM MES, pH 6, 100 mM lithium acetate, 15% (w/v) polyethylene glycol PEG3350). All crystals grew in 1–2 days and they were washed in cryoprotectant (40% PEG3350 in precipitant solution plus additives, see below) and flash frozen in liquid nitrogen for data collection. Crystallization and cryoprotection were performed as follows: (1) LIG1$^{WT}$ 3′ddC 200 mM Mg$^{2+}$ complex: nicked 18mer was assembled by annealing oligos 1, 2, and 3 (Supplementary Table 1), and the cryoprotectant was supplemented with 200 mM MgCl$_2$. (2) LIG1$^{WT}$ 3′ddC 2 mM Mg$^{2+}$ complex: nicked 18mer was assembled by annealing oligos 1, 2, and 3 (Supplementary Table 1), and the cryoprotectant was supplemented with 2 mM MgCl$_2$. (3) LIG1$^{E346A/E592A}$ 3′ddC 200 mM Mg$^{2+}$ complex: nicked 18mer was assembled by annealing oligos 1, 2, and 3 (Supplementary Table 1), and the cryoprotectant was supplemented with 200 mM MgCl$_2$. (4) LIG1$^{WT}$ 3′OH EDTA complex: nicked 18mer was assembled by annealing oligos 1, 3, and 4 (Supplementary Table 1), and the cryoprotectant was supplemented with 2 mM EDTA. (5) LIG1$^{E346A/E592A}$ 3′OH EDTA complex: nicked 18mer from annealing oligos 1, 3, and 4 (Supplementary Table 1), and the cryoprotectant was supplemented with 2 mM EDTA. (6) LIG1$^{E346A/E592A}$ 3′–8oxoG EDTA complex: nicked 18mer was assembled by annealing oligos 3, 5, and 6 (Supplementary Table 1), and the cryoprotectant was supplemented with 2 mM EDTA. (7) LIG1$^{E592R}$ 3′OH EDTA complex: nicked 18mer from annealing oligos 1, 3, and 4 (Supplementary Table 1), and the cryoprotectant was supplemented with 2 mM EDTA. X-ray diffraction data was collected on Beamline 22-ID of the Advanced Photon Source at a wavelength of 1.000 Å. X-ray diffraction data were processed and scaled using the HKL2000 suite[36]. All structures were solved by molecular replacement using PDB entry 1X9N as a search model with PHASER[37]. Iterative rounds of model building in COOT and refinement with PHENIX[38] were used to produce the final models.

**Gel-based ligation assay**. Oligonucleotides substrates were purchased from IDT, Midland Certified Reagent Company, or the Keck Center at Yale University and purified by denaturing PAGE[30]. Annealing was performed in an annealing buffer containing 10 mM NaMES, pH 6.5 and 50 mM NaCl. The 28-mer-C:G oligo was formed by annealing oligos 7, 8, and 9 (Supplementary Table 1). The 8oxoG-containing substrates were formed by annealing oligos 8 and 10 with either oligo 11 (28-mer-8oxoG:C) or oligo 12 (28-mer-8oxoG:A). For purification of 8oxoG-containing oligonucleotides, 10 mM 2-mercaptoethanol was included in the loading buffer and gel. Ligation assays were carried out at 37 °C in a standard reaction buffer consisting of 50 mM NaMOPS at pH 7.5, 1 mM dithiothreitol, 0.05 mg/mL BSA, and sufficient NaCl to maintain a constant ionic strength of 150 mM. Concentrations of ATP, MgCl$_2$, 28-mer substrate, and LIG1 were varied as indicated below. For reactions stated to be at physiological conditions, final concentrations of 1 mM ATP and 2 mM MgCl$_2$ were used. The free magnesium concentration for these reactions was calculated to be 1 mM using the dissociation constants for the Mg$^{2+}$•ATP complex[30]. Unless noted otherwise, reactions were quenched in the standard loading buffer (50 mM EDTA/90% formamide/0.01% xylene cyanol/ 0.01% bromophenol blue). Quenched samples were heated to 95 °C and separated on either a 15 or 20% (w/v) polyacrylamide gel containing 8 or 6.6 M urea, respectively. The fluorescein-labeled oligonucleotide was detected using an Amersham Typhoon 5 imager (GE), and gel images were analyzed using Image-Quant TL software (GE). Rate constants are reported as the average ± standard deviation for at least three independent experiments.

**Single-turnover ligation assays**. To determine the individual rates of the adenylyl transfer and nick-sealing steps, single-turnover reactions were performed with 800 nM Δ232 LIG1 and 80 nM DNA substrate. Reactions to determine Mg$^{2+}$ dependence of the chemical steps were performed in a KinTek RQF-3 quench-flow apparatus, where LIG1$^{WT}$ or LIG1$^{E346A/E592A}$ was loaded in a separate sample loop from the 28-mer-C:G substrate, both at double the desired final concentrations to account for dilution upon mixing. Both ligase and DNA solutions were prepared with the standard reaction buffer supplemented with desired concentrations of MgCl$_2$. Drive syringes were loaded with the standard reaction buffer and reactions were quenched using a solution of 1 N H$_2$SO$_4$ and 0.25% SDS. Upon quenching, 20 μL of the quenched sample was immediately neutralized in a solution containing 300 mM Tris base, 15 mM EDTA and 0.5 × loading buffer.

Reactions used to measure the rates of ligation for the 28-mer 8oxoG:A substrate were also performed in the RQF-3 but were quenched with the standard quench solution. Reactions containing the 28-mer-8oxoG:C substrate were carried out in a heat-block set at 37 °C and quenched with the standard quench solution. Rates were obtained from fits to the single-turnover data using a two-step irreversible published Berkeley–Madonna model[30].

**Determining the steady-state dependences of ligation**. Steady-state dependences were performed using the standard reaction buffer and 0.1–10 nM LIG1$^{WT}$ or LIG1$^{E346A/E592A}$. For both the Mg$^{2+}$ and ATP dependences, 500 nM of the 28-mer substrate was used along with either 200 μM ATP or 20 mM MgCl$_2$, respectively. For the DNA substrate dependences at saturating levels of MgCl$_2$ and ATP,

reactions contained 20 mM MgCl$_2$ and 200 μM ATP. DNA substrate dependences were also performed under physiological conditions. For reactions with low concentrations of the DNA substrate (<5 nM), reactions were quenched with a 50 mM EDTA solution and completely dried down in a speed-vac before being resuspended in a minimal volume of the standard loading buffer. Initial rates for all steady-state reactions were determined by linear regression, and the rates were fit with the Michaelis–Menten equation (Eq. 1).

$$\frac{V_{init}}{[E]} = \frac{V_{max}[S]}{K_M + [S]} \quad (1)$$

For reactions in which an accumulation of both intermediate and product was observed, the initial rates of total substrate disappearance were used. The fraction of observed abortive ligation events was determined using Eq. 2, in which V is the initial rate for formation of intermediate or product.

$$\text{Fraction abortive ligation} = \frac{(V_{intermediate})}{(V_{product} + V_{intermediate})} \quad (2)$$

**Determining the effect of APTX on ligation**. Both steady-state and single-turnover reactions for measuring the effect of APTX on ligation were carried out under physiological ATP and Mg$^{2+}$ concentrations. Steady-state reactions contained 0 or 10 nM APTX, 1 nM LIG1$^{WT}$, or LIG1$^{E346A/E592A}$, and 500 nM nicked DNA substrate. Using initial rates (V) for total substrate disappearance in the presence and absence of APTX, the percentage of suppressed ligation was calculated using Eq. 3.

$$\text{Suppressed ligation (\%)} = \frac{V_{-APTX} - V_{+APTX}}{V_{-APTX}} \times 100\% \quad (3)$$

Single-turnover reactions were initiated by 1:1 mixing of a solution containing 1600 nM LIG1$^{WT}$ or LIG1$^{E346A/E592A}$, 160 nM DNA substrate and 0.1 mM EDTA with a separate solution containing 0 or 1600 nM H260N APTX and 4 mM MgCl$_2$. Both the ligase and APTX solutions contained the standard reaction buffer supplemented with 1 mM ATP. Reactions were quenched as stated above for single-turnover reactions. Data were fit in Berkeley–Madonna to a modified two-step model, wherein LIG1 partially dissociates and is displaced from the adenylylated intermediate by H260N APTX (Fig. 4c and Supplementary Fig. 9). The rates determined from the fits were used to calculate the catalytic commitment factor as shown in Eq. 4:

$$\text{Catalytic commitment} = \frac{k_{seal}}{k_{seal} + k_{off}} \quad (4)$$

**Reporting summary**. Further information on research design is available in the Nature Research Reporting Summary linked to this article.

## Data availability

A reporting summary for this article is available as a Supplementary Information file. Atomic coordinates and structure factors for the reported crystal structure have been deposited with the RCSB Protein Data Bank under accession numbers 6P09, 6P0A, 6P0B, 6P0C, 6P0D, 6P0E, and 6Q1V. The source data underlying all main text Figs. (3a, 3b, 3c, 3d, 3e, 4a, 4b, 4d, 4e, 4f, 6a), Supplementary Figures (1a,1b, 1c, 2a, 2b, 3a, 3b, 3c, 4a, 4b, 4c, 6a, 6b, 7a, 7b, 7c, 7d, 10a, 10b, 10c, 10d, 10e, and Supplementary Tables (2–4), are provided as a Source Data file. The data supporting the findings of this study are available from the corresponding authors upon reasonable request.

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

## Acknowledgements

The research was supported by the US National Institute of Health Intramural Program, US National Institute of Environmental Health Sciences (NIEHS) 1Z01ES102765 (to R.S. W), and the National Institutes of Health Extramural Program, National Institute of General Medical Studies (NIGMS) R01GM130763 (to P.J.O.). T.J.J. was supported in part by a training fellowship from the Cellular Biotechnology Training Program at the University of Michigan (T32GM130763). We thank L. Pedersen of the NIEHS Collaborative crystallography group for data collection support and the Advanced Photon Source (APS) Southeast Regional Collaborative Access Team (SER-CAT) for beamline access. Use of the APS was supported by the US Department of Energy, Office of Science, Office of Basic Energy Sciences, under Contract No. W-31–109-Eng-38. We thank Dr. Tom Kunkel and Dr. Lars Pedersen for comments on the manuscript.

## Author contributions

Conceptualization R.S.W., P.T., T.J.J., P.J.O., methodology P.T., M.J.S., T.J.J., P.J.O., R.S.W.; investigation P.T., M.J.S., T.J.J., A.R.R,; writing—original draft, R.S.W., P.J.O., T.J. J., P.T.; writing—reviewing and editing, R.S.W., P.J.O., T.J.J., P.T., A.R., M.J.S.; funding acquisition R.S.W., P.J.O.; supervision R.S.W., P.J.O.

## Competing interests

The authors declare no competing interests.
