## [Peer Review File · Nature Communications]

Reviewers' comments:

Reviewer #1 (Remarks to the Author):

"Two-Tiered Enforcement of High-Fidelity DNA Ligation"

The manuscript presents a crystallographic analysis of human DNA Ligase 1, and biochemical analysis of human DNA Ligase 1 and aprataxin. The focus of the study is a DNA substrate with an 8-oxo-G nucleotide on the 3 prime end of a DNA break. The reported structures are of significantly higher resolution than that of the previously reported structure of human ligase 1, and thus provide more detailed views of the architecture of DNA ligase 1 bound to a DNA break. In particular, specific Mg²⁺ binding sites were identified, and the study focuses on the role of a metal binding site that they term a high fidelity site. This name stems from the fact that mutation of two Glu residues that coordinate metal binding leads to an overall increase in ligation efficiency of DNA substrates containing an 8-oxo-G nucleotide on the 3 prime side of the DNA break. The loss of these residues and metal binding is proposed to lead to less stringent structural requirements for the 3' end of the break, presumably allowing the mutant version of DNA ligase to remain suitably engaged for ligation for a longer period of time than wild-type on 8-oxo-G containing substrates. The biochemical data clearly show that there is no major change in the ligation chemistry steps between the wild-type and mutant protein, but rather a difference in substrate utilization reflected in a large difference in Km for DNA substrate. The experiments are convincing and the study provides new insights into DNA ligase engagement of DNA substrates.

Specific comments:

The mutation of two glutamates to alanines introduces more available volume around the DNA 3' end by significantly shortening the side chain of two residues. So in addition to removing the metal, the double mutant itself could be removing structural restrictions on binding to the 3' end. It is difficult therefore to attribute the gain of function phenotype solely to the loss of Mg²⁺ binding. Making this distinction clear is important in the context of discussing Ligase 3 and Ligase 4 - they might impose the same "volume" restriction on the 3'end, but perhaps not using a metal binding site. Do the published Ligase 3 and Ligase 4 structures fill this same area with other residues?

There is not a clear rationale given for the placement of the 8-oxo-G on the 3' end, as opposed to the penultimate nucleotide or elsewhere on the 3' end. Since it seems that polymerases can extend beyond 8-oxo-G, the nucleotide could conceivably end up at other positions around the break. Is there an 8-oxo-G position where even the double mutant will not accommodate the structural perturbation?

Since some of the complexes were crystallized bound to ligation competent DNA substrates (bearing a 3' OH), it would be interesting to know if the ligation reaction proceeds with the addition of Mg²⁺, and whether structures could be determined for these product complexes. With the slower kinetics of the double mutant ligase bound to 8-oxo-G, it might be possible to capture a meaningful snapshot of the organization of the active site and the role of the catalytic metal proposed in this study.

Minor comments:

In panel C of the S1 legend, this seems like an error: "... with the addition of 500 nM LIG1 nicked DNA."

The plots of the biochemical reactions are quite small.

S6 legend references Figure 5B, but Figure 4B seems to be the appropriate figures.

First sentence of last paragraph of Results/Discussion (page 12): Breiba et al Embo J (2004) should be included in the references to DNA polymerases with 8oxoG:A base pairs.

Page 13, "Although this hypothesis that is yet to be tested..." Needs re-wording.

Reviewer #2 (Remarks to the Author):

The manuscript "Two-Tiered Enforcement of High-Fidelity DNA Ligation" by Tumbale et al the authors provide penetrating insights into the molecular mechanism of error-free DNA ligation by human DNA ligase 1, by high resolution X-ray structures of DNA ligase I-complexes in combination with functional characterization of the enzyme in activity studies. They show that fidelity of the ligation reaction is ensured at two steps of the ligation reaction, the DNA-ampylation (step 2) and nick-sealing (step 3) which is mediated by the stringent coordination geometry enforced by a magnesium ion. Here mismatches and oxidative DNA lesions lead to an alteration of the alignment between the DNA substrate at the enzymes active site, leading to abortive ligation reaction. Such stalled ligation events are resolved by aprataxin (APTX) an enzyme which catalyzes the release of adenylate groups covalently linked to 5'-phosphate termini. Moreover they show that mutation of the Mg²⁺ coordinating residues leads to a reduced fidelity and increase in ligation of erroneous substrates. The manuscript is of very high quality, well written, the experiments are well designed and the data provided fully support the conclusions drawn.

Thus I have only one point, that in my opinion, should be addressed prior to publication: Mutations to Ala could also impact on the structural flexibility. Hence effects might not be entirely due to the fact that the chemical properties of an aa side chain is lacking, but a combinatorial effect, including alteration of the flexibility. The authors discuss that the stringent geometry of the Mg²⁺ hexa-coordination in the HF-site strongly impacts on the ligation fidelity. As pointed out effects observed by disruption of the Mg²⁺ binding site through mutation of the coordinating residues could have multiple causes and might not solely caused by loss of the ion. In the introduction the authors make compare ligases and DNA polymerases, the latter being a very prominent example of how small alteration in the chemical structures of nucleobases are communicated by the stringent Mg-coordination regulating enzyme activity. For polymerases it is well known that by replacing the Mg-ion by manganese, which allows a relaxed coordination geometry, leads to a higher error rate during replication. Thus what is happening if Mn²⁺ is added to the ligation reaction? Does that allow ligation of substrates with mismatches and oxidative lesions?

Other minor comments:

The authors say that

Are there any crystal contacts made by the DNA that could impact on the structural arrangement?

page 3 bottom: introduce abbreviation MTH1

Tumbale, Jurkiw et al, response to referees.

We thank the referees for their positive and constructive comments on this work. We have addressed the critiques in full and with additional data and discussion where appropriate. Please find the referee comments below in *blue italics* with our responses and data below in black text.

Reviewer #1 (Remarks to the Author): "Two-Tiered Enforcement of High-Fidelity DNA Ligation"

The manuscript presents a crystallographic analysis of human DNA Ligase 1, and biochemical analysis of human DNA Ligase 1 and aprataxin. The focus of the study is a DNA substrate with an 8-oxo-G nucleotide on the 3 prime end of a DNA break. The reported structures are of significantly higher resolution than that of the previously reported structure of human ligase 1, and thus provide more detailed views of the architecture of DNA ligase 1 bound to a DNA break. In particular, specific Mg²⁺ binding sites were identified, and the study focuses on the role of a metal binding site that they term a high-fidelity site. This name stems from the fact that mutation of two Glu residues that coordinate metal binding leads to an overall increase in ligation efficiency of DNA substrates containing an 8-oxo-G nucleotide on the 3 prime side of the DNA break. The loss of these residues and metal binding is proposed to lead to less stringent structural requirements for the 3' end of the break, presumably allowing the mutant version of DNA ligase to remain suitably engaged for ligation for a longer period of time than wild-type on 8-oxo-G containing substrates. The biochemical data clearly show that there is no major change in the ligation chemistry steps between the wild-type and mutant protein, but rather a difference in substrate utilization reflected in a large difference in Km for DNA substrate. The experiments are convincing and the study provides new insights into DNA ligase engagement of DNA substrates.

Specific comments:

The mutation of two glutamates to alanines introduces more available volume around the DNA 3' end by significantly shortening the side chain of two residues. So in addition to removing the metal, the double mutant itself could be removing structural restrictions on binding to the 3' end. It is difficult therefore to attribute the gain of function phenotype solely to the loss of Mg²⁺ binding.

Thank you, this is a very important point. As highlighted by the referee, our X-ray structures of the E346A/E592A mutant reveal a cavity that facilitates dynamic protein-DNA binding in the mutant ligase structure. We hypothesize that the impaired fidelity of the $LIG1^{E346A/E592A}$ enzyme results from relaxation of the strict geometric requirements for Mg^{2+} metal coordination during DNA binding, and creation of this cavity. To more directly dissect the contribution of metal coordination to ligation fidelity, we generated an additional E592R mutation and assessed ligation and abortive ligation profiles on undamaged and 8oxoG containing substrates. Similar to the $LIG1^{E346A/E592A}$ mutant, $LIG1^{E592R}$ displays markedly attenuated abortive ligation (Figure 6A), and enhanced nick sealing activity (Supplementary figure 10) on 8oxoG:A and 8oxoG:C substrates. We further determined the crystal structure of $LIG1^{E592R}$ at 1.85 Å (Figure 6B and Table S1). In this structure, the R592 substitution occupies the approximate position of E592 plus the HiFi magnesium ligand, and forms a salt bridge to the phosphodiester backbone in place of the metal-DNA contact. Compared to $LIG1^{WT}$ a small cavity is formed, but the interface is typified by a high complementary of the protein and 3' DNA terminus. This cavity is reduced relative to that which is present in $LIG1^{E346A/E592A}$, but very similar effects were observed on the fidelity of DNA ligation (Figures 6A and 6C). Thus, substitution of a direct protein-DNA interface for the native protein-metal-DNA interface is not sufficient for high-fidelity ligation. We conclude that high fidelity of $LIG1$ is dictated by DNA binding by distant regions of the protein which converge on the nucleic acid backbone at the nexus of the 3' strand. Metal coordination between the DBD, AdD and DNA imparts $LIG1$ with a precision measuring tool that is tuned in part by the strict requirements for coordinating magnesium between the phosphodiester backbone and the HiFi metal ligands E346 and E592.

Making this distinction clear is important in the context of discussing Ligase 3 and Ligase 4 - they might impose the same "volume" restriction on the 3'-end, but perhaps not using a metal binding site. Do the published Ligase 3 and Ligase 4 structures fill this same area with other residues?

Thank you, this is great point. We compared the 3' strand binding properties of LIG1 with LIG3 (PDB 3L2P) and LIG4 (PDB 6BKF). LIG1 exhibits robust surface complementarity that extends from the nick through the N-4 position (Figure 6D). While the protein-DNA contacts of LIG3 and LIG4 are mediated by topologically equivalent DNA binding loops, these loops adopt different structures in the three DNA ligases (Figures 6D and 6E). The LIG3 and LIG4 protein-DNA interface are comparatively discontinuous and are interrupted with cavities, resembling the LIG1^{E346A/E592A} and LIG1^{E592R} low fidelity mutants. We speculate that discontinuous 3' strand binding contributes to the relaxed fidelity for 3' mismatches observed for human LIG3 (ref 38), and the mutagenic DNA repair reactions performed during DSB repair by LIG4 (refs 10, 11).

There is not a clear rationale given for the placement of the 8-oxo-G on the 3' end, as opposed to the penultimate nucleotide or elsewhere on the 3' end. Since it seems that polymerases can extend beyond 8-oxo-G, the nucleotide could conceivably end up at other positions around the break. Is there an 8-oxo-G position where even the double mutant will not accommodate the structural perturbation?

Thanks for this point. This work was an extension of seminal observations of LIG1 poisoning by 8oxoG incorporation during 1nt gap filling base excision repair reactions (see Caglayan et al 2017). We clarify this in the text.

Since some of the complexes were crystallized bound to ligation competent DNA substrates (bearing a 3' OH), it would be interesting to know if the ligation reaction proceeds with the addition of Mg²⁺, and whether structures could be determined for these product complexes. With the slower kinetics of the double mutant ligase bound to 8-oxo-G, it might be possible to capture a meaningful snapshot of the organization of the active site and the role of the catalytic metal proposed in this study.

Thank you for this great suggestion. We have attempted to initiate the ligation reaction in LIG1-3'OH DNA crystals by adding Mg²⁺. Unfortunately, the Mg²⁺ soaked crystals in the present crystal form and growth conditions did not diffract under the conditions used. Efforts are underway to stabilize these crystals for so-called time-lapse experiments.

Minor comments:

In panel C of the S1 legend, this seems like an error: "... with the addition of 500 nM LIG1 nicked DNA."

Yes, you are correct. We have changed this to "500 nM nicked DNA".

The plots of the biochemical reactions are quite small.

Ok, we have tried to maximize text/plot visibility within the journal guidelines.

S6 legend references Figure 5B, but Figure 4B seems to be the appropriate figures.

Ok, thank you. We have corrected this issue.

First sentence of last paragraph of Results/Discussion (page 12): Breiba et al Embo J (2004) should be included in the references to DNA polymerases with 8oxoG:A base pairs.

Ok that is true, and we have added this reference.

Page 13, "Although this hypothesis that is yet to be tested..." Needs re-wording.

Ok, thank you, we have reworded this sentence to "Although this hypothesis is yet to be tested, the physiological roles of APTX are complex as this protein interacts with other proteins related to the DNA damage response, including XRCC1 and XRCC4³⁴."

Reviewer #2 (Remarks to the Author):

The manuscript "Two-Tiered Enforcement of High-Fidelity DNA Ligation" by Tumbale et al the authors provide penetrating insights into the molecular mechanism of error-free DNA ligation by human DNA ligase 1, by high resolution X-ray structures of DNA ligase I-complexes in combination with functional characterization of the enzyme in activity studies. They show that fidelity of the ligation reaction is ensured at two steps of the ligation reaction, the DNA-ampylation (step 2) and nick-sealing (step 3) which is mediated by the stringent coordination geometry enforced by a magnesium ion. Here mismatches and oxidative DNA lesions lead to an alteration of the alignment between the DNA substrate at the enzymes active site, leading to abortive ligation reaction. Such stalled ligation events are resolved by aprataxin (APTX) an enzyme which catalyzes the release of adenylate groups covalently linked to 5'-phosphate termini. Moreover they show that mutation of the Mg²⁺ coordinating residues leads to a reduced fidelity and increase in ligation of erroneous substrates. The manuscript is of very high quality, well written, the experiments are well designed and the data provided fully support the conclusions drawn.

Thus I have only one point, that in my opinion, should be addressed prior to publication: Mutations to Ala could also impact on the structural flexibility. Hence effects might not be entirely due to the fact that the chemical properties of an aa side chain is lacking, but a combinatorial effect, including alteration of the flexibility. The authors discuss that the stringent geometry of the Mg²⁺ hexa-coordination in the HF-site strongly impacts on the ligation fidelity. As pointed out effects observed by disruption of the Mg²⁺ binding

site through mutation of the coordinating residues could have multiple causes and might not solely caused by loss of the ion. In the introduction the authors make compare ligases and DNA polymerases, the latter being a very prominent example of how small alteration in the chemical structures of nucleobases are communicated by the stringent Mg-coordination regulating enzyme activity. For polymerases it is well known that by replacing the Mg-ion by manganese, which allows a relaxed coordination geometry, leads to a higher error rate during replication. Thus what is happening if Mn²⁺ is added to the ligation reaction? Does that allow ligation of substrates with mismatches and oxidative lesions?

We thank the referee for this interesting suggestion. First, we created another mutant, E592R, to investigate the effects of complete ablation of the HiFi site with the double-alanine mutations (see response to reviewer 1). The new structure and new biochemical data support the model that Mg²⁺ at the HiFi site confers high fidelity ligation activity on LIG1 (See new Figure 6). Second, we analyzed LIG1 activity in multiple turnover ligation reactions while titrating either Mg²⁺ or Mn²⁺ cofactors (Figure R1). Notably, the effects of Mn²⁺ are complex, with Mn²⁺ stimulating LIG1 activity (k_{cat}) at low mM concentration of Mn²⁺ and inhibiting at higher concentrations. This observation is in stark contrast to the hyperbolic dose responses to Mg²⁺ titration under similar conditions. Clearly the effects of Mn²⁺ do not fit a single site model. While we share this reviewer's idea that Mn²⁺ could potentially impact fidelity by occupying the HiFi site, we feel that this line of inquiry will require extensive experimentation that is beyond the scope of the present manuscript and is therefore appropriate for future work.

[REDACTED]

Other minor comments: The authors say that

Are there any crystal contacts made by the DNA that could impact on the structural arrangement?

The core of the ligase active site is somewhat shielded from the more distant crystal contacts and we don't have any reason to hypothesize that crystal contacts might affect the specific comparisons that we highlight in this work.

page 3 bottom: introduce abbreviation MTH1

Thank you, we have added "MutT homolog 1" to the text.

REVIEWERS' COMMENTS:

Reviewer #1 (Remarks to the Author):

In my opinion, the authors have addressed the key concerns raised in the first review. The addition of the E592R mutant structure nicely adds to the comparison of human ligases 1, 3, and 4.

Reviewer #2 (Remarks to the Author):

The authors have addressed all points raised. I recommend the manuscript for publication by Nature Communication.